# Analysis of the Validity of Perioperative Antibiotic Prophylaxis in Maxillofacial Surgery

**DOI:** 10.3390/jcm11195812

**Published:** 2022-09-30

**Authors:** Iwona Niedzielska, Marcin Kotowski, Anna Mertas, Michał Bąk, Szczepan Barnaś, Damian Niedzielski

**Affiliations:** 1Department of Cranio-Maxillo-Facial Surgery, Medical University of Silesia in Katowice, ul. Francuska 20-24, 40-027 Katowice, Poland; 2Department of Microbiology and Immunology, Medical University of Silesia in Katowice, ul. Jordana 19, 41-808 Zabrze, Poland; 3Department of Otorhinolaryngology, 4th Military Clinical Hospital, ul. Weigla 5, 50-981 Wrocław, Poland

**Keywords:** antibiotic therapy, perioperative antibiotic prophylaxis, surgical site infection, exogenous infections, endogenous infections, oral physiological flora

## Abstract

Perioperative antibiotic prophylaxis is the standard in surgical departments. The type of operation, the duration of the procedure, the degree of microbiological purity of the operating field and the current clinical condition of the patient determine its administration. The aim of this study was to validate the antibiotic prophylaxis used in a Maxillofacial Surgery Department for a group of trauma and non-trauma patients. To that end, an observational prospective cohort study was carried out. The study was conducted on a group of 83 patients of the Department of Cranio-Maxillo-Facial Surgery who were divided into a group of trauma patients (*n* = 43) and one of non-trauma patients (*n* = 40). In both groups, the classic microbiological tests were carried out, and the results were analyzed in relation to: the study group, age, sex, duration of surgery, type of surgical access. Most bacterial strains were isolated at the initial stage of the operation. Gram (+) cocci were isolated more often in the trauma group and Gram (−) rods in the non-trauma group. Significantly more often, strains of fungi were noted in the initial stage of the procedure in the trauma group. We conclude that the use of perioperative antibiotic prophylaxis in the Maxillofacial Surgery Departments is justified.

## 1. Introduction

Perioperative antibiotic prophylaxis (PAP) allows us to reduce the incidence of surgical site infection (SSI) in every field of surgery, provided it is properly administered [1,2]. However, it should be remembered that PAP is not an attempt to sterilize tissues and does not replace the correct preparation of the patient for the procedure—antiseptics and well-conducted surgical techniques are still critical. Plus, the effectiveness of antibiotics depends on the species of the microorganism, the degree of drug penetration into the tissues, the patient’s immune status, the local epidemiological situation of the hospital and surgical ward and the type of surgery and its duration [3,4,5].

It is currently believed that oral procedures are associated with a high risk of transient bacteremia. Bacteremia is estimated to occur in 84–100% of extractions, 52% of single extractions and removal of supragingival calculus deposits, 43% of periodontal pocket depth measurements and 31% of endodontic procedures. The probability of bacteriemia after brushing teeth and irrigation is about 40% [6].

According to the opinion of the Polish board (the National Antibiotic Protection Program) regarding the use of antibiotics in dentistry, surgeries in the maxillary area associated with the highest risk of bacteremia are: bone decortication and tooth extraction [1,2]. Similarly, other procedures performed in this anatomical region are burdened with a relatively high risk of bacteremia. These include orthognathic surgery, maxillary sinus surgery, surgical treatment of jaw fractures and enucleation of bone cysts [1,7].

In a healthy population, bacteremia does not cause any complications due to well-functioning immune processes. However, in the case of illnesses or damaged structures (e.g., heart, kidney), bacteremia may lead to systemic infection [8].

A very high percentage (almost 90%) of post-extraction bacteremia has been confirmed in patients without PAP, which seems to be the final and irrefutable argument for the need for chemo-prophylaxis in patients with a high or medium risk of systemic infection [3]. However, doubts apply to patients with a low risk of systemic infection, or those who do not present any type of burden. The main factors that have an impact on pathogenic flora in maxillofacial surgery have still not been fully explained. The possible factors are the type of surgery, the choice of materials and implants, the connection of the external (extraoral) and internal (intraoral) environments, contact with the periodontal gap during the procedure, the level of disinfection of the operating field and finally, systemic conditions [9,10,11,12,13].

The latest recommendations of the Polish Dental Association and the National Antibiotic Protection Program for dentistry for 2016–2020 provide guidelines for PAP for specific procedures in the field of dentistry, mostly based on an assessment of the risk of complications with antibiotic therapy in relation to non-recipients. PAP administration is considered appropriate in non-immunocompetent patients, and no PAP is recommended for dentistry procedures for immunocompetent patients. In those guidelines, PAP is recommended in terms of the dose, time of administration and type of antibiotic.

While exploring the literature on which those recommendations were based, we noted that conclusions were rarely drawn based on bacteriological studies of the surgical site combined with postoperative PAP efficacy. Table 1 presents the details of the studies that were referenced by the recommendations of the Polish Dental Association and the National Antibiotic Protection Program for dentistry for 2016–2020. Since the legitimacy of PAP in the era of accredited units could not be subject to discussion in the undertaken analysis, it was not possible to assess the indications for PAP, only its legitimacy. An attempt was made to evaluate PAP depending on the duration of the procedure, the time taken to reach the wound (external, internal) and the type of surgery (trauma, other). The aim of this study was to analyze the validity of the principles of antibiotic prophylaxis used for patients at the Maxillofacial Surgery Department, divided into trauma and non-trauma patients.

## 2. Materials and Methods

The study was conducted from 1 March 2017 to 1 March 2018 on a group of 83 patients who underwent surgical treatment at the Department of Cranio-Maxillo-Facial Surgery of the Medical University of Silesia in Katowice. The patients were divided into two groups depending on their diagnosis:

a group of trauma patients undergoing osteosynthesis of facial cranial fractures (n = 43);a group of non-trauma patients who underwent a procedure other than osteosynthesis of fractures (n = 40).

Minors and patients with a medium or high risk of systemic infection were excluded from the study.

Before surgery, outside the operating room, patients had complete removal of oral infection, were given oral hygiene instructions and used mouthwash with antiseptic (15 mL of 0.2% chlorhexidine and a 30 s rinse in accordance with Tomás et al.’s findings [21]), which was repeated in the operating room.

In accordance with generally accepted guidelines for perioperative antibiotic therapy, patients of both groups received cefazoline and metronidazole intravenously [22,23]. Cefazolin was administered in doses of 1.0 g (patient weight < 80 kg) or 2.0 g (patient weight > 80 kg) 30 min before incision of the skin or mucosa, with the next dose of 1.0 g intravenous cefazoline after 4 h. Metronidazole at a dose of 15 mg/kg of the body mass was administered for 30–60 min to complete the infusion one hour before the procedure, and postoperatively at a dose of 7.5 mg/kg of the body mass after 6–12 h. In cases of hypersensitivity to beta-lactam antibiotics, clindamycin was administered at 900 mg intravenously to complete the infusion 30 min before the procedure. The infusion time was between 30 and 40 min [1].

For both groups of patients, material for microbiological tests was collected by swabbing at two time points: (1) at the beginning of the surgery just after the incision of the tissues, and (2) at the end of the surgery just before closing of the wound (from its bottom). The collected swabs were placed in a suitable transport medium and delivered to the Central Laboratory of the Independent Public Clinical Hospital of the Silesian Medical University, where microbiological tests were carried out. The time from collecting the material to its delivery for testing did not exceed 24 h. The consent of the Bioethical Committee operating at the Silesian Medical University in Katowice was confirmed by resolution no. KNW/022/KB1/15/17. Wound healing was reported and complaints reported, as well as the clinical status in subsequent medical checks for up to a month after surgery.

Microbiological tests were carried out using classic methods used in microbiological diagnostics. The material collected from 83 patients was seeded on appropriate culture media, to multiply and isolate pure microbial cultures. The drug susceptibility of isolated bacterial strains was determined by the disk diffusion method and E-tests. The collected material was seeded on suitable culture media to multiply and isolate pure microbial cultures, i.e., aerobic bacteria were cultivated on Columbia agar solid medium with 5% sheep blood at 37 °C; anaerobic bacteria were cultivated on Schaedler K3 solid medium with 5% sheep blood at 37 °C; Candida fungi were cultivated and pre-identified using the Biomerieux ChromID Candida chromogenic medium. After isolation and multiplication of cultured microbial strains, their species identification was carried out using the Biomerieux Vitek 2 Compact device and the Mikrobionet 2.0 computer program, and by a set of biochemical tests (companies: BIOMERIEUX Catalase, Slidex Staph Kit and API Candida). The implementation of the test and interpretation of the results obtained were in accordance with the current EUCAST (European Committee on Antibiotic Susceptibility Testing) recommendations [24] and the recommendations of the National Center for Drug Susceptibility Microbes (KORLD) [2].

## 3. Results

The obtained results of microbiological tests were analyzed in relation to: age, sex, type of surgery (assignment to group I or II), duration of surgery (up to 60 min or over 60 min) and access to the surgical field (extra- and/or intraoral). Finally, the results of 43 trauma patients and 40 non-trauma patients were compared. Patients undergoing procedures other than osteosynthesis of fractures of the facial bones of the skull were undergoing surgery of the maxillary sinuses (n = 9), cystic bone changes (n = 16), benign tumors of soft tissues or facial bones (n = 8) or salivary glands (n = 7). In the group of trauma patients (43 patients), 19 osteosynthesis of fractures were operated on in less than 60 min, while in 24 patients, the procedure lasted over 60 min. Osteosynthesis was performed using intraoral access in 20 cases, extraoral in 14 cases and combined intra- and extraoral access in 9 cases. Respectively, in a group of 40 non-trauma patients, 23 non-osteosynthesis procedures were performed within 60 min, and 17 patients underwent surgery lasting more than 60 min. In non-trauma patients, intraoral access was gained in 31 cases, extraoral in 2 cases and combined intra- and extraoral access in 7 cases. The characteristics of both groups are presented in Table 2.

No disturbing signs of local or general inflammation were noted in any patient. Localized reactions in the form of edema, disappearing pain or slight redness of the wound lasting up to 10 days after surgery were considered typical in the course of healing.

Most bacterial strains, mainly Gram (+) bacteria, were isolated at the beginning of the surgical procedure from trauma patients, followed by the patients who underwent surgery other than osteosynthesis, also at the beginning of the surgical procedure. Less often, but also in significant amounts, Gram (−) rods were isolated, often when the procedure was not associated with injury, with the numbers higher compared to the trauma group. Significantly more often, strains of fungi were noted at the initial stage of osteosynthesis compared to the final stage and/or non-trauma patients, where no fungi were found in the initial stage of surgery, with the exception of one patient (Table 3).

Among Gram (+) cocci, *Staphylococcus epidermidis* was most often isolated (more often from the bottom of the wound before its closure compared to the beginning of the procedure and the non-trauma group), *Streptococcus mitis* (more often in the non-trauma than trauma group and more often in the initial stage of surgery than the final one for both groups), *Streptococcus parasanguinis* (more often in the trauma than non-trauma group and comparably for both phases of the procedure) and *Streptococcus viridans* (more often from the bottom of the wound shortly after the incision of the tissues in both groups compared to the later stage of the procedure) (Table 4). *Staphylococcus capitis* was found only in the group of trauma patients, more often at the beginning than the end of the procedure. *Neisseria* spp. was more frequently isolated among Gram (−) cocci at the beginning of the procedure compared to the later stages of the procedure in both groups of patients. *Enterobacter cloacae* was isolated at the beginning of the procedure in non-trauma patients compared to trauma patients who did not have this species. *Candida albicans* was identified only in the group of trauma patients, more often in wounds shortly after tissue incision than before suturing (Table 4).

Seventeen species of microorganisms were isolated in the final stage of surgery in the trauma group and the initial in the non-trauma group. However, as many as 21 species of microorganisms were found at the beginning of osteosynthesis and just before wound closure during surgery other than osteosynthesis (Table 5).

When we analyzed the relationship between the time of surgical procedure, we found that in the case of longer-lasting (more than 60 min) osteosynthesis (group of trauma patients), more strains of microorganisms (n = 26) were isolated than in shorter procedures (n = 22). On the contrary, in the second—non-trauma—group of patients, more strains were isolated in shorter (n = 24) than in longer procedures (n = 23). Most often, they were Gram (+) cocci (Table 6).

The influence of the duration of the procedure on the type of microorganism grown in the operating field is presented in Table 6. *Staphylococcus epidermidis MSCNS* was more often isolated in longer-lasting osteosynthesis procedures than in shorter procedures. On the contrary, in the non-trauma group, more strains of this bacterial species appeared in shorter procedures than in longer procedures. *Escherichia coli* occurred in the group of trauma patients in the case of shorter osteosynthesis, and in the group of non-trauma patients during longer surgery. One case of *Candida crusei* was reported in a longer procedure in the non-trauma group (Table 7).

Most microbial species were found in cultures of material taken from the wound when the procedure was longer than 60 min in a group of patients without injuries to the facial part of the skull. Thirteen bacterial species were noted for shorter procedures of both groups, and most often they were Gram (+) cocci (Table 8).

Most strains of microorganisms, mainly Gram (+) cocci, were isolated in the case of procedures performed with intraoral access regardless of the procedure, less often with both approaches and the least frequently with intraoral access. Gram (−) rod strains appeared in the trauma group only in the case of oral access, and in cases of intraoral and combined intra- and extraoral access in the non-trauma group (Table 9).

In the trauma group with intraoral access, the following Gram (+) cocci were isolated: *Streptococcus anginosus*, *Streptococcus constellatus*, *Streptococcus mitis*, *Streptococcus sanguinis*, *Streptococcus salivarius* and *Streptococcus virans*. Those were not present in the extraoral or combined intra- and extraoral access subgroup. Intraoral surgical intervention also determined the isolation of other species that did not appear when selecting other approaches, such as *Escherichia coli*, *Haemophilus influenzae* and *Candida albicans*. Most sterile cultures were found in the trauma group by selecting extraoral access (Table 10).

Most species of microorganisms were isolated when the intraoral route was selected in the non-trauma group in the Maxillofacial Surgery Department. *Staphylococcus aureus MSSA*, *Staphylococcus hominis*, *Streptococcus agalactiae*, *Streptococcus anginosus*, *Streptococcus constellatus*, *Streptococcus gordoni*, *Streptococcus parasanguinis*, *Streptococcus salivarius*, *Moraxella*, *Infusa*, *Enterobacter cloacae* and *Pseudomonas aeruginosa* were present then (Table 11).

The largest number of microbial species (19 species) was found in the group of non-trauma patients with intraoral access; fewer species were found in the trauma group with intraoral access (13 species). No microbial species were isolated when taking extraoral access for non-trauma patients (Table 12).

In this research, examples of microorganisms in the surgical site were observed that were resistant not only to PAP (cefazoline and metronidazole) but also to empirical therapy with first- or second-line antibiotics (amoxicillin/clavulanic acid, clindamycin) for the treatment of perimaxillary infections according to the Polish guidelines [1].


**Example 1:**


At the beginning of the procedure, *Enterobacter cloacae* was isolated from material collected from a patient in a non-trauma group, in which, according to EUCAST guidelines, the use of cefazoline and metronidazole in prophylaxis is not recommended. *Pseudomonas aeruginosa* was grown from the material collected at the end of the procedure. In accordance with EUCAST guidelines, for this microorganism, it is not recommended to use PAP with cefazoline and metronidazole, or amoxicillin with clavulanic acid—a drug used in the empirical therapy for a previously diagnosed *Enterobacter cloacae* infection.


**Example 2:**


In a patient from the trauma group, a bacterium belonging to the *Haemophilus influenzae* species was isolated from the material taken at the beginning and at the end of the procedure, for which, according to EUCAST guidelines, the use of cefazoline and metronidazole in prophylaxis, as well as clindamycin (used in this patient in empirical therapy), is not recommended.

These examples indicate the possible benefit of performing microbiological tests, the results of which allow the use of targeted therapy that effectively prevents complications of the surgical procedure, in particular, generalized infections originating from the surgical site. However, when analyzing the drug-sensitiveness of the bacterial species isolated from the maxillofacial surgery patients in the study, except for in two patients, it was found that the antibiotic prophylaxis used was consistent with the drug-sensitiveness profile of the isolated microorganisms.

## 4. Discussion

The oldest and most extensive analysis, comprising the observation of 63,000 clean, contaminated and dirty wounds, showed that the greatest risk of complications occurs in dirty, open and old wounds, reaching 40% of cases [25]. Referring to the above, it can be assumed that an extended time of surgery should play an important role in the bacterial contamination of the wound. Plus, the older wounds were associated with higher bacterial loads, which confirms the results of our analysis i.e., more bacterial species were isolated in the final stage of surgery in trauma patients. In the case of longer-lasting osteosynthesis (over an hour), more strains of microorganisms (n = 26) were isolated than in a shorter time (n = 22), and compared to the second non-trauma group of patients, more strains were isolated in a shorter time (n = 24) than the longer duration of the procedure (23 strains). Most often, they were Gram (+) strains. Therefore, McPhee and Papadakis [26], as well as Perdikaris and Pefanis [27] justify the use of prolonged antibiotic therapy when the surgery is prolonged, which significantly affects the growth of wound-contaminating microorganisms. In the presented research, there was only a slight difference in isolated strains depending on the duration of surgery.

In another analysis, the time elapsed between the administration of cefazoline or other short-acting cephalosporin and the surgical incision was clarified on the basis of 4472 cases. Thus, the time of antibiotic administration in the PAP was clarified [28]. According to the Scottish Intercollegiate Guidelines Network’s antibiotic prophylaxis guidelines (2008), for most treatments where prevention is required, one dose of antibiotic is sufficient; in some situations, prevention may be extended to 24 h. However, extension of perioperative prophylaxis over 24 h, according to the authors, does not reduce the risk of infectious complications; instead, it may increase the risk of drug resistance and side effects [29]. In 2006, the Surgical Infection Society drew attention to a small number of studies mostly published over 30 years previously on prophylactic antibiotic use in orthopedics. Two scientific societies believe that in open fractures, not only should PAP be used but also antibiotics should be continued. Hauser et al. recommended prophylactic antibiotic use [30]. The results of the conducted tests prove the above-mentioned since in the extraoral connection to the surgical wound, no pathogenic strains were found at its bottom, while the intraoral access was conducive to the appearance of many different types of bacterial strains and even fungi. The mixed approach also disturbed the bacterial balance, suggesting the need for PAP taking both approaches. This would also be in line with the latest directives of the Working Group of the Polish Dental Association and the National Antibiotic Protection Program, which propose that orthopedic treatment of condyloma fractures or an extraoral operative approach does not require PAP. The society does not recommend the routine use of antibiotic prophylaxis in immunocompetent patients, and it advises that the decision to implement antibiotic prophylaxis should be balanced [1].

In head and neck surgery, the insertion of dental implants alone is not recommended, but the simultaneous insertion of bone grafts for implantation already requires PAP. According to this society’s guidelines, PAP should be given for maxillary sinus surgery, nasal cavities, large cysts and jaw tumors, as well as in orthognathic surgery, in which the entrance to the sinus or nasal cavity is made, along with bone resection, free lobe surgery or pediatric operations on the lymphatic system of the neck, in which they are connected to the respiratory tract, and in bone grafts. By exploring many publications cited by the authors of this working group, and those included in the table in the introduction to this article, we gained the impression that indications for PAP are methodologically unjustified. In the methodology, the authors rely on very small research groups [17] or only on clinical observations of PAP’s success, or on narrow research groups of very different surgical procedures. Wound cultures were collected only in individual studies [18]. The authors carried out the research in two groups: with PAP (PAP + autogenous bone graft + implant) and a control (without PAP + autogenous bone graft + implant), isolating *Streptococcus mitis* (test group 38.8%; control group 31.2%), *Streptococcus acidominimus* (test group 33.3%; control group 31.2%), *Streptococcus uberis* (test group 22.2%; control group 18.7%) and *Streptococcus morbillorum* (test group 16.6%; control group 18.7%). They found that PAP reduces but does not eliminate the infection [18].

The results of our research lead us to the following conclusions. One must be very careful with the rigorous approach taken to antibiotic prophylaxis. An individual approach to the problem should be taken and the possible complications should not be underestimated. The analysis showed that the isolated strains were susceptible to the recommended PAP, but two cases of fairly serious infections that broke out of the PAP positive-interaction pattern warn us that caution must be taken with the schematic approach to antibiotic prophylaxis. An interesting aspect of future research could be a microbiological analysis of a larger size of research material, with the option of disabling PAP, and research developing indications for its use based on such studies, not just clinical observations.

## 5. Conclusions

Perioperative antibiotic prophylaxis (PAP) with cefazoline and metronidazole in patients treated for injuries and undergoing selected surgical procedures in Maxillofacial Surgery Departments is justified in terms of bacterial sensitivity. Only in two patients (of 83) were the isolated strains resistant to the PAP administered.

## Figures and Tables

**Table 1 jcm-11-05812-t001:** Summary of findings.

Authors	Material and Methods	Conclusions
Abu-Ta’a, “Adjunctive Systemic Antimicrobial Therapy vs. Asepsis in Conjunction with Guided Tissue Regeneration” [14]	40 patients: 20 with PAP; 20 without PAP DFDBA procedure: bone allograft Clinical observation for lack of complications	No benefits of PAP
Funahara et al., “Prevention of Surgical Site Infection after Oral Cancer Surgery by Topical Tetracycline” [15]	Research group *n* = 61 (administration of tetracycline ointment to the back of the tongue every 6 h for 48 h after surgery for oral cancer); control group *n* = 56 Multifactorial analysis and symptoms, and swabs	Regional use of tetracycline as an effective way to prevent wound infection after oral cancer surgery
Arteagoitia et al., “Amoxicillin/Clavulanic Acid 2000/125 Mg to Prevent Complications Due to Infection Following Completely Bone-Impacted Lower Third Molar Removal” [5]	Research group *n* = 58, single administration of APO before extraction of third molar tooth; placebo *n* = 60, Clinical observation for lack of complications	No benefits of PAP
Bortoluzzi et al., “A Single Dose of Amoxicillin and Dexamethasone for Prevention of Postoperative Complications in Third Molar Surgery” [16]	Group of 50 patients with extracted third molar teeth Group 1 (G1), prophylactic dose of 2 g amoxicillin and 8 mg dexamethasone Group 2 (G2), prophylactic dose of 2 g amoxicillin and 8 mg placebo Group 3 (G3), prophylactic dose of 8 mg dexamethasone and 2 g placebo Group 4 (G4), placebo Clinical observation for lack of complications	No benefits of PAP
Lindeboom and van den Akker, “A Prospective Placebo-Controlled Double-Blind Trial of Antibiotic Prophylaxis in Intraoral Bone Grafting Procedures” [17]	Group of 20 patients Prevention *n* = 10 Placebo *n* = 10 Intraoral bone grafts Clinical observation up to 3 months	Efficiency of PAP confirmed
Mauceri et al., “The Role of Antibiotic Prophylaxis in Reducing Bacterial Contamination of Autologous Bone Graft Collected from Implant Site” [18]	34 patients: 18 with PAP (1 gr. amoxicillin + clavulanic acid; 12 h and 1 h before surgery) 16 without PAP 15 days before the procedure, oral hygiene session and instruction, and 0.2% chlorhexidine mouthrinse twice a day; A surgical swab was taken and then the strains found were evaluated	The tested antibiotic prophylaxis regimen reduces but does not eliminate the risk of infection
Chiesa-Estomba et al., “Antibiotic Prophylaxis in Clean Head and Neck Surgery” [19]	Retrospective SSI assessment Without prevention, *n* = 77 Antibiotic prophylaxis, *n* = 109 Resection of the submandibular gland, parotid gland resection, cystic mandibular resection	A prophylactic antibiotic is not necessary for clean, gentle head and neck surgery
Danda and Ravi, “Effectiveness of Postoperative Antibiotics in Orthognathic Surgery” [20]	Meta-analysis of five clinical trials involving 532 patients undergoing orthognathic surgery. Wound infection occurred in 30 of 268 patients in the short-term prophylaxis group (frequency, 11.2%) and in 10 of 264 patients in the prolonged treatment group (frequency 3.8%)	According to the authors, extended antibiotic therapy was more effective in reducing the risk of postoperative wound infection, but they stressed that more research is needed to harmonize the appropriate regimen
Mauceri et al., “The Role of Antibiotic Prophylaxis in Reducing Bacterial Contamination of Autologous Bone Graft Collected from Implant Site” [18]	Patients with PAP in autologous transplants around the implant Research group—15 days before the procedure, given instructions for rinsing the mouth with chlorhexidine and PAP Control group without PAP Cultures were demonstrated for specific strains in both groups	PAP reduces but does not eliminate the infection

**Table 2 jcm-11-05812-t002:** Characteristic of examined patients.

Parameter		Trauma Patients	Non-Trauma Patients
Procedure duration	up to 60 min	19	23
	above 60 min	24	17
Sex	female	4	21
	male	29	19
Access to surgical field	extraoral	14	2
	intraoral	20	31
	intra- and extraoral	9	7

**Table 3 jcm-11-05812-t003:** Number of strains of microorganisms isolated from the material taken from the surgical site at the beginning and immediately after the surgical procedure in trauma and non-trauma patients.

Group of Microorganisms	Trauma Patients *n* = 43		Non-Trauma Patients *n* = 40	
	Beginning of Procedure	End of Procedure	Beginning of Procedure	End of Procedure
Gram (+) cocci	54	42	39	37
Gram (−) cocci	1	1	5	2
Gram (+) rods	0	0	1	0
Gram (−) rods	5	2	7	7
Fungi	8	3	0	1
IN TOTAL:	68	48	52	47

**Table 4 jcm-11-05812-t004:** List of microorganisms isolated from material taken from the surgical site at the beginning and immediately after the surgery performed on trauma and non-trauma patients.

Species of Microorganisms	Trauma Patients *n* = 43		Non-Trauma Patients *n* = 40	
	Beginning of Procedure	End of Procedure	Beginning of Procedure	End of Procedure
Gram (+) cocci:				
*Enterococcus faecalis*	1	0	0	0
*Enterococcus saccharolyticus*	1	0	0	0
*Lactococcus garvieae*	1	0	0	0
*Staphylococcus aureus MSSA*	0	0	0	1
*Staphylococcus capitis*	3	2	0	0
*Staphylococcus hominis*	1	0	0	1
*Staphylococcus epidermidis MSCNS*	14	17	12	7
*Staphylococcus lentus*	1	1	0	0
*Staphylococcus saprophyticus*	0	0	1	0
*Streptococcus agalactiae*	0	1	0	1
*Streptococcus anginosus*	3	2	0	4
*Streptococcus constellatus*	0	1	0	3
*Streptococcus gordoni*	0	0	0	1
*Streptococcus mitis*	7	4	9	3
*Streptococcus mutans*	1	1	1	0
*Streptococcus sanguinis*	2	2	4	3
*Streptococcus parasanguinis*	7	7	3	6
*Streptococcus pneumonia*	0	0	0	1
*Streptococcus pseudoporcinus*	0	0	1	0
*Streptococcus salivarius*	2	2	2	3
*Streptococcus thoraltensis*	0	0	1	0
*Streptococcus viridans*	5	1	5	3
*Streptococcus vestibularis*	0	1	0	0
Gram (−)cocci:				
*Moraxella catarrhalis*	0	0	1	1
*Neisseria* spp.	1	1	4	1
Gram (+) rods:				
*Rothia dentocariosa*	0	0	1	0
Gram (−) rods:				
*Acinetobacter baumanii*	1	0	0	1
*Enterobacter cloacae*	0	0	2	1
*Escherichia coli*	1	1	2	2
*Haemophilus influenza*	1	1	2	2
*Haemophilus parainfluenzae*	0	0	1	0
*Klebsiella pneumonia*	1	0	0	0
*Pseudomonas aeruginosa*	1	0	0	1
Fungi:				
*Candida albicans*	8	3	0	0
*Candida crusei*	0	0	0	1
Negative cultures:	5	9	5	8

**Table 5 jcm-11-05812-t005:** Number of species of microorganisms isolated from material collected from the surgical site at the beginning and immediately after the surgery performed on trauma and non-trauma patients.

Groups of Microorganisms	Trauma Patients *n* = 43		Non-Trauma Patients *n* = 40	
	Beginning of Procedure	End of Procedure	Beginning of Procedure	End of Procedure
Gram (+) cocci	14	13	10	13
Gram (−) cocci	1	1	2	2
Gram (+) rods	0	0	1	0
Gram (−) rods	5	2	4	5
Fungi	1	1	0	1
IN TOTAL:	21	17	17	21

**Table 6 jcm-11-05812-t006:** Number of strains of microorganisms isolated from material collected from the surgical site immediately after the end of surgery performed on trauma and non-trauma patients divided into groups depending on the duration of the surgery.

Groups of Microorganisms	Trauma Patients *n* = 43		Non-Trauma Patients *n* = 40	
	Procedure Lasting Up to 60 min *n* = 19	Procedure Lasting over 60 min *n* = 24	Procedure Lasting Up to 60 min *n* = 23	Procedure Lasting over 60 min *n* = 17
Gram (+) cocci	18	24	20	17
Gram (−) cocci	1	0	1	1
Gram (−) rods	1	1	3	4
Fungi	2	1	0	1
IN TOTAL:	22	26	24	23

**Table 7 jcm-11-05812-t007:** Number of strains of microorganisms isolated from material collected from the surgical site immediately after the end of surgery performed on trauma and non-trauma patients divided into groups depending on the duration of the surgery.

Type of Microorganisms	Trauma Patients *n* = 43		Non-Trauma Patients *n* = 40	
	Procedure Lasting Up to 60 min *n* = 19	Procedure Lasting over 60 min *n* = 24	Procedure Lasting Up to 60 min *n* = 23	Procedure Lasting over 60 min *n* = 17
Gram (+) cocci:				
*Staphylococcus aureus MSSA*	0	0	0	1
*Staphylococcus capitis*	1	1	0	0
*Staphylococcus hominis*	0	0	0	1
*Staphylococcus epidermidis MSCNS*	6	11	4	3
*Staphylococcus lentus*	0	1	0	0
*Streptococcus agalactiae*	1	0	1	0
*Streptococcus anginosus*	0	2	2	2
*Streptococcus constellatus*	1	0	2	1
*Streptococcus gordoni*	0	0	1	0
*Streptococcus mitis*	2	2	1	2
*Streptococcus mutans*	1	0	0	0
*Streptococcus sanguinis*	1	1	2	1
*Streptococcus parasanguinis*	3	4	4	2
*Streptococcus pneumonia*	0	0	1	0
*Streptococcus salivarius*	0	2	2	1
*Streptococcus viridans*	1	0	0	3
*Streptococcus vestibularis*	1	0	0	0
Gram (−) cocci:				
*Moraxella catarrhalis*	0	0	0	1
*Neisseria* spp.	1	0	1	0
Gram (−) rods:				
*Acinetobacter baumanii*	0	0	1	0
*Enterobacter cloacae*	0	0	0	1
*Escherichia coli*	1	0	0	2
*Haemophilus influenza*	0	1	2	0
*Pseudomonas aeruginosa*	0	0	0	1
Fungi:				
*Candida albicans*	2	1	0	0
*Candida crusei*	0	0	0	1
Negative cultures:	5	4	6	2

**Table 8 jcm-11-05812-t008:** Number of species of microorganisms isolated from material collected from the surgical site immediately after the end of surgery performed on trauma and non-trauma patients divided into groups depending on the duration of the surgery.

Groups of Microorganisms	Trauma Patients *n* = 43		Non-Trauma Patients *n* = 40	
	Procedure Lasting Up to 60 min *n* = 19	Procedure Lasting over 60 min *n* = 24	Procedure Lasting Up to 60 min *n* = 23	Procedure Lasting over 60 min *n* = 17
Gram (+) cocci	10	8	10	10
Gram (−) cocci	1	0	1	1
Gram (−) rods	1	1	2	3
Fungi	1	1	0	1
IN TOTAL:	13	10	13	15

**Table 9 jcm-11-05812-t009:** Number of strains of microorganisms isolated from material collected from the surgical site immediately after the end of surgery performed on trauma and non-trauma patients divided into groups depending on the method/access route of the surgical procedure.

Groups of Microorganisms	Trauma Patients *n* = 43			Non-Trauma Patients *n* = 40		
	Procedure with Intraoral Access *n* = 20	Procedure with Extraoral Access *n* = 14	Procedure with Intra- and Extraoral Access *n* = 9	Procedure with Intraoral Access *n* = 31	Procedure with Extraoral Access *n* = 2	Procedure with Intra- and Extraoral Access *n* = 7
Gram (+) cocci	21	9	11	33	0	4
Gram (−) cocci	1	0	0	2	0	0
Gram (−) rods	2	0	0	3	0	3
Fungi	3	0	0	1	0	0
IT TOTAL:	27	9	11	39	0	7

**Table 10 jcm-11-05812-t010:** List of microorganisms isolated from material collected from the surgical site immediately after the end of surgery performed in trauma patients divided into groups depending on the method/access route for the surgery.

Types of Microorganisms	Trauma Patients *n* = 43		
	Procedure with Intraoral Access *n* = 20	Procedure with Extraoral Access *n* = 14	Procedure with Intra- and Extraoral Access *n* = 9
Gram (+) cocci:			
*Staphylococcus capitis*	0	1	1
*Staphylococcus epidermidis MSCNS*	4	5	8
*Staphylococcus lentus*	0	0	1
*Streptococcus agalactiae*	0	1	0
*Streptococcus anginosus*	2	0	0
*Streptococcus constellatus*	1	0	0
*Streptococcus mitis*	4	0	0
*Streptococcus mutans*	1	0	0
*Streptococcus sanguinis*	2	0	0
*Streptococcus parasanguinis*	5	1	1
*Streptococcus salivarius*	2	0	0
*Streptococcus viridans*	1	0	0
*Streptococcus vestibularis*	0	1	0
Gram (−) cocci:			
*Neisseria* spp.	1	0	0
Gram (−) rods:			
*Escherichia coli*	1	0	0
*Haemophilus influenzae*	1	0	0
Fungi:			
*Candida albicans*	3	0	0
Negative cultures:	3	6	0

**Table 11 jcm-11-05812-t011:** List of microorganisms isolated from material collected from the surgical site immediately after the completion of surgery performed in non-trauma patients divided into groups depending on the method/access route for surgery.

Types of Microorganisms	Non-Trauma Patients *n* = 40		
	Procedure with Intraoral Access *n* = 31	Procedure with Extraoral Access *n* = 2	Procedure with Intra- and Extraoral Access *n* = 7
Gram (+) cocci:			
*Staphylococcus aureus MSSA*	1	0	0
*Staphylococcus hominis*	1	0	0
*Staphylococcus epidermidis MSCNS*	5	0	2
*Streptococcus agalactiae*	1	0	0
*Streptococcus anginosus*	4	0	0
*Streptococcus constellatus*	3	0	0
*Streptococcus gordoni*	1	0	0
*Streptococcus mitis*	2	0	1
*Streptococcus sanguinis*	3	0	0
*Streptococcus parasanguinis*	6	0	0
*Streptococcus pneumoniae*	1	0	0
*Streptococcus salivarius*	3	0	0
*Streptococcus viridans*	2	0	1
Gram (−) cocci:			
*Moraxella catarrhalis*	1	0	0
*Neisseria* spp.	1	0	0
Gram (−) rods:			
*Acinetobacter baumanii*	1	0	0
*Enterobacter cloacae*	0	0	1
*Escherichia coli*	1	0	1
*Haemophilus influenzae*	1	0	0
*Pseudomonas aeruginosa*	0	0	1
Fungi:			
*Candida crusei*	1	0	0
Negative cultures:	5	2	1

**Table 12 jcm-11-05812-t012:** Number of species of microorganisms isolated from material collected from the surgical site immediately after the completion of surgery performed on trauma and non-trauma patients divided into groups depending on the method/access route of the surgery.

Types of Microorganisms	Trauma Patients *n* = 43			Non-Trauma Patients *n* = 40		
	Procedure with Intraoral Access *n* = 20	Procedure with Extraoral Access *n* = 14	Procedure with Intra- and Extraoral Access *n* = 9	Procedure with Intraoral Access *n* = 31	Procedure with Extraoral Access *n* = 2	Procedure with Intra- and Extraoral Access *n* = 7
Gram (+) cocci	9	5	4	13	0	3
Gram (−) cocci	1	0	0	2	0	0
Gram (−) rods	2	0	0	3	0	3
Fungi	1	0	0	1	0	0
IN TOTAL:	13	5	4	19	0	6

## Data Availability

The data presented in this study are available on request from the corresponding authors. Publicly data sharing is not applicable to this article due to privacy policy.

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
