# Peer review of "Analysis of the Validity of Perioperative Antibiotic Prophylaxis in Maxillofacial Surgery"

_jcm, 2022, doi:10.3390/jcm11195812_

Round 1
Reviewer 1 Report
You paid attention to a lot of variables so the article might seem a bit confusing. In the complex is a good work, I really liked the way you used to evaluate all the aspects.
There are a few dated references, but I can understand that this is a very discussed topic in the past and then slightly underestimated, while now it is making a comeback, so it's not simple to find recent valid articles.
- line 67 the following sentence is unclear "PAP administration was considered appropriate in the group of non-immunocompetent patients, and dentistry procedures were recommended in the group of immunocompetent patients, in which PAP is recommended in terms of dose, time of administration and type of antibiotic."
Author Response
- line 67 the following sentence is unclear "PAP administration was considered appropriate in the group of non-immunocompetent patients, and dentistry procedures were recommended in the group of immunocompetent patients, in which PAP is recommended in terms of dose, time of administration and type of antibiotic."
Thank you very much for this remark. The sentence indeed is confusing and has been rewritten.

Reviewer 2 Report
It is a good work that brings us closer, despite the fact that there are some methodological issues that should be considered to what type of microbiota is in patients before and after surgical treatment and based on a certain protocol. But they do not allow conclusions to be drawn, beyond these data.
Introduction
-As reflected in the introduction, in reference 8, more value should be given to bacteriemias produced by multiple procedures in healthy patients
-In Table 1, we do not know how those items were chosen. Also because those and not others. They also do not look correctly referenced
Methodology
-Which antiseptic was used for the rinses and for how long.
-Why metronidazole + cefazolin, I understand that based on 2018 protocol, but could justify it
-How was the sample collected: saliva, points of paper, from where, etc.?
-The bibliographical references do not seem well ordered, after the [8], the [11] follows and I do not see the [10], for example.
- It would be nice to have confirmation from the ethics committee.
Results
-What does it mean that they mainly isolated, were the others isolated or not?
-Weren't three samples taken?
….“For both groups of patients, material for microbiological tests was collected at the beginning of the surgery, just after the incision of the tissues and at the end of the surgery but just before closing the wound (from its bottom)” ….
-What percentage of strains were isolated?
-They talk about empiric therapy with amoxicillin + clavulanic acid / clindamycin, and we have not seen it in the methodology
-We do not understand the meaning of the examples, in results?
Discussion
-The references [10 and 11] must be placed in their respective author. And it only makes sense to quote them, if compared to the results obtained, if they should not be considered in the introduction.
Conclusions
They seem contradictory and it does not seem that they can be extracted from the study.
All patients received prophylaxis, what would have happened if they did not receive it?
Author Response
It is a good work that brings us closer, despite the fact that there are some methodological issues that should be considered to what type of microbiota is in patients before and after surgical treatment and based on a certain protocol. But they do not allow conclusions to be drawn, beyond these data.
Introduction
-As reflected in the introduction, in reference 8, more value should be given to bacteriemias produced by multiple procedures in healthy patients
Thank you very much for this remark. We have emphasized this in lines 39-43, however if you consider this not enough we will extend this section.
-In Table 1, we do not know how those items were chosen. Also because those and not others. They also do not look correctly referenced
Thank you very much for this point. The citations were reformatted. As for the selection criteria – the Table 1 lists the studies that were referenced by the recommendations of the Polish Dental Association and the National Antibiotic Protection Program for dentistry for 2016-2020. It is now better elucidated in the text.
Methodology
-Which antiseptic was used for the rinses and for how long.
In accordance with Tomás et al. findings the 15ml of 0,2% chlorhexidine for 30 seconds were used.
-Why metronidazole + cefazolin, I understand that based on 2018 protocol, but could justify it
The administration of cefazoline with metronidazole was justified by the polish guidelines. Most recent meta-analysis (Iocca et al., “Antibiotic Prophylaxis in Head and Neck Cancer Surgery.”) supports the administration of cefazoline and metronidazole in Head and Neck Surgery.
-How was the sample collected: saliva, points of paper, from where, etc.?
The material for microbiological sampling was collected by swabbing the bottom of the wound
-The bibliographical references do not seem well ordered, after the [8], the [11] follows and I do not see the [10], for example.
This has been fixed. Thank you for this remark.
- It would be nice to have confirmation from the ethics committee.
In lines 112-115 there is a confirmation of the Bioethical Committee approval.
Results
-What does it mean that they mainly isolated, were the others isolated or not?
Thank you for your remark – we assume that it concerned lines 158-160 – the sentence has been rewritten to be clear to the reader.
-Weren't three samples taken?
The methods section has been rewritten to stop confusion: “For both groups of patients, material for microbiological tests was collected by swabbing at two time points: 1) at the beginning of the surgery just after the incision of the tissues and 2) at the end of the surgery just before closing of the wound (from its bottom)”. The samples were collected at two time points.
-What percentage of strains were isolated?
Thank you for your question - for the isolated strains we are reporting on absolute numbers.
-They talk about empiric therapy with amoxicillin + clavulanic acid / clindamycin, and we have not seen it in the methodology
Thank you for this note – the sentence in lines 256-259 has been rewritten so it is clearer. The point is that in the course of research the strains were isolated which were resistant both to antibiotics used for PAP as well as first and second line antibiotics in perimaxillary infections.
-We do not understand the meaning of the examples, in results?
Thank you for this note – the sentence in lines 256-259 has been rewritten so it is clearer. The point is that in the course of research the strains were isolated which were resistant both to antibiotics used for PAP as well as first and second line antibiotics in perimaxillary infections. The idea of presentint the examples is to emphasize the need for sensitivity testing.
Discussion
-The references [10 and 11] must be placed in their respective author. And it only makes sense to quote them, if compared to the results obtained, if they should not be considered in the introduction.
The references have been rearranged and the compoarision to the results have been included.
Conclusions
They seem contradictory and it does not seem that they can be extracted from the study.
The conclusions have been rewritten to make it clearer and bolder.
All patients received prophylaxis, what would have happened if they did not receive it?
That is interesting question, however the applicable guidelines in our hospital setting made it impossible to perform such a research.

Reviewer 3 Report
Corrections of grammatical and spelling errors are needed.
Author Response
Corrections of grammatical and spelling errors are needed.
Thank you for your remark – the corrections have been made.

Round 2
Reviewer 2 Report
Revision 2
Estimado editor/autor, el paper ha mejorado sustanciamene, pero no deja de ser una pressentaci´ón de datos obtenidos, sin un comparación clara y creo que deben solventarse difentes aspectos que obligan a una revisión mayor.
1.-The whole paragraph of the introduction is surprising, without any bibliographical reference, and the only one is from 2004:
„Very high percentage (almost 90%) of post-extraction bacteremia have been 54 confirmed in patients without PAP, which seems to be the final and irrefutable argument 55 for the need for chemo-prophylaxis in patients with high and medium risk of systemic 56 infection. [3] However, doubts will apply to patients with low risk of systemic infection, 57 those who do not present any type of burden. It has still not been fully explained what are 58 the main factors that have an impact on pathogenic flora in maxillofacial surgery. The 59 possible factors are the type of surgery, the choice of materials and implants, the 60 connection of the external (extraoral) and internal (intraoral) environment, contact with 61 the periodontal gap during the procedure, the level of disinfection of the operating field 62 and finally – systemic conditions.”
2.- -Justify why in Table 1 the data of the author Maurice et al, are not grouped.
3.- What is the objective of the work and its justification?
4.- Methodology
They still do not justify the protocol used, which could have some application in complicated fractures, but in no way in other less serious procedures. In any case, if the rest of the issues are resolved, we can accept it, as long as they justify that it is the protocol of your hospital. Think that in addition, it is a review of cancer patients and among its results it specifies:
“There was no significant difference between clindamycin and no antibiotic (OR: 2.3, 95%CrI: 0.59-9.9). Clindamycin plus aminoglycoside seemed to give a slight protection from SSI compared to clindamycin alone (OR: 0.30, 95%CrI: 0.09-0.99) or no antibiotic (OR: 0.13, 95%CrI: 0.02-067).”
5.- Results
We still do not understand the meaning of the examples, if only in two cases the sensitivity was not correct, it seems clear that it could be done empirically, right?:
“Analyzing drug-sensitiveness of bacterial species isolated from the maxillofacial surgery patients qualified for the study, except for two patients, it was found that antibiotic prophylaxis used was consistent with the drug-sensitiveness profile of isolated microorganisms.”
6.- Discussion
The discussion seems somewhat clearer, but the paragraph:
“It would also be in line with the latest directives of the Working Group of the Polish Dental Association and the National Antibiotic Protection Program, which believes that orthopedic treatment of condyloma fractures or extra-oral operative approach does not require PAP.”
Must be re-referenced
7.- Conclusions
The work does not allow justifying the type of prophylaxis used, it can be concluded that with this prophylaxis "x" events have occurred, but no more
8.- There are references that do not seem appropriate in terms of their format, either because some information is missing or it is incomplete:
2, 3,6,7,17,19,21,22 and 24
Sincerely, in Barcelona, on 25-7-22
